# MicroRNAs Which Can Prognosticate Aggressiveness of Bladder Cancer

**DOI:** 10.3390/cancers11101551

**Published:** 2019-10-14

**Authors:** Edyta Marta Borkowska, Tomasz Konecki, Michał Pietrusiński, Maciej Borowiec, Zbigniew Jabłonowski

**Affiliations:** 1Medical University of Lodz, Chair of Laboratory and Clinical Genetics Department of Clinical Genetics, 251 Pomorska Street, 92-213 Lodz, Poland; Michal.pietrusinski@umed.lodz.pl (M.P.); maciej.borowiec@umed.lodz.pl (M.B.); 2Medical University of Lodz, 1st Clinic of Urology, 113 Żeromski Street, 90-549 Lodz, Poland; Tomasz.konecki@umed.lodz.pl (T.K.); zbigniew.jablonowski@umed.lodz.pl (Z.J.)

**Keywords:** Bladder cancer, microRNA, genetic marker, progression

## Abstract

Bladder cancer (BC) is still characterized by a very high death rate in patients with this disease. One of the reasons for this is the lack of adequate markers which could help determine the biological potential of the tumor to develop into its invasive stage. It has been found that some microRNAs (miRNAs) correlate with disease progression. The purpose of this study was to identify which miRNAs can accurately predict the presence of BC and can differentiate low grade (LG) tumors from high grade (HG) tumors. The study included 55 patients with diagnosed bladder cancer and 30 persons belonging to the control group. The expression of seven selected miRNAs was estimated with the real-time PCR technique according to miR-103-5p (for the normalization of the results). Receiver operating characteristics (ROC) curves and the area under the curve (AUC) were used to evaluate the feasibility of using selected markers as biomarkers for detecting BC and discriminating non-muscle invasive BC (NMIBC) from muscle invasive BC (MIBC). For HG tumors, the relevant classifiers are miR-205-5p and miR-20a-5p, whereas miR-205-5p and miR-182-5p are for LG (AUC = 0.964 and AUC = 0.992, respectively). NMIBC patients with LG disease are characterized by significantly higher miR-130b-3p expression values compared to patients in HG tumors.

## 1. Introduction

Bladder cancer (BC) is characterized by the high rate of non-muscle invasive BC (NMIBC) at the moment of diagnosis (75–80%) [1,2]. Transitional cell carcinoma (TCC) constitutes the majority of the urothelial carcinoma of the bladder. There are two described alternative molecular pathways of developing BC, characterized by different genetic changes and different biological potentials. The first alternative includes changes of papillary and an always non-invasive character, while the other alternative can be either papillary or non-papillary and is often invasive (into the lamina propria—T1 stage; or muscularis propria—T2 stage) [3,4]. Patients suffering from muscle invasive BC (MIBC) at the moment of the initial diagnosis are treated with radical cystectomy (RC). This is not the optimal solution, as patients’ quality of life after RC is low and a high rate of relapse and death has been observed within a short period of time after operation [5,6]. As far as patients with NMIBC are concerned, it is impossible to predict which of them will have disease progression. In consequence, they undergo systematic cystoscopy examinations aimed at assessing the disease development stage. This also decreases patients’ quality of life (time in hospital, stress, uncertainty connected with another examination) and generates enormous costs for the health care [7].

MicroRNAs (miRNAs) are known to be dysregulated in bladder cancer (BC) and implicated in the pathogenesis of the development of bladder tumors mostly via their influence on genes involved in two molecular pathways, specifically the gene which codes fibroblast growth factor receptor 3 (FGFR3) and the gene which codes tumor protein 53 (TP53). Numerous miRNA studies have identified histological grade and stage (pT) classification-dependent miRNA expression and have proven the existence of miRNAs alterations related to the two divergent pathways found in the development of NMIBC and MIBC [8,9]. Only a few studies have analyzed miRNA as a prognostic and predictive biomarker [10,11,12]. Each miRNA can have multiple targets, and changes in their expression profile could have a magnified effect on cellular phenotype. The previously published studies emphasize the possible prognostic potential of some miRNAs to predict progression and disease specific or overall survival in BC patients. Unfortunately, none of these miRNAs are used in routine practice. This in the result of quite a few factors: Using different platforms for assessing marker expressions, using various biological samples (tissue or cell lines) secured in different ways (paraffin, RNAlater, freezing), and using various normalization methods and reference genes [13]. Some analyses are based on relative expression and others are based on absolute expression. Finally, these factors also include the lack of a control group. That is why we decided to evaluate the expression of selected miRNAs in an adequately selected group of both NMIBC and MIBC patients characterized by the high rate of observed progression.

In tumors, downregulated miRNAs are considered to be tumor suppressor candidates, whereas miRNAs with increased expression may play a promotional role in cancer progression. Potential BC suppressors include miR-100, miR-99a, miR-202, and miR-30a. Some miRNAs, including miR-145-5p (locus on chromosome 5), miR-195, and miR-199a-5p have been shown to inhibit the proliferation of or induce the apoptosis of BC cells [14]. MiR-145-5p appears to play a key role as a tumor suppressor by targeting N-cadherin and its downstream effector matrix metalloproteinase-9 (MMP9), and it is the most frequently reported downregulated miRNA in BC. MiR-205-5p (locus on chromosome 1), miR-182-5p (locus on chromosome 7), mir-130b-3p (locus on chromosome 22), miR-10a-5p and miR-21-5p (loci on chromosome 17), and miR-20a-5p (locus on chromosome 13) are mainly overexpressed in BC tissue. They promote proliferation, migration, and invasion, and they inhibit BC cells apoptosis. The potential target/regulator for miR-130b-3p and miR-205-5p is the *PTEN* gene (phosphatase and tensin homolog); for miR-182-5p, it is the *SMAD4* gene (drosophila protein, mothers against decapentaplegic homolog 4); and for miR-10a-5p, it is the *FGFR3* gene [15]. miR-21-5p overexpression is related to *TP53* inactivation, invasion, and tumor progression. It has been seen to be simultaneously upregulated in the tissue, plasma and urinary exosomes of BC patients, but its role needs further elucidation. However, there are still conflicting results regarding the function of miRNAs in publications, so, for our analysis, we chose a panel of the best described miRNAs for BC and the miRNAs connected with genes or chromosomes whose genetic alterations are well documented in pathogenesis BC [14,15].

## 2. Materials and Methods

The tested group consisted of tumor tissue samples stored in the tissue bank in the Clinical Genetics Department, the Chair of Clinical and Laboratory Genetics, Medical University of Lodz. The tumor tissues were obtained during the TURBT (transurethral resection of bladder tumor) examination at the Urology Ward of the University Clinical Hospital Military Memorial Medical Academy in Lodz. Official permission to conduct the tests was granted by the Bioethics Advisory Commission at Lodz Medical University, No. RNN/62/15/ KE/M, and the patients signed consent forms. The tumors selected for RNA isolation were submerged in an RNA later solution (Sigma) and stored at −20 °C before isolation time. The tests were carried out on a group of 55 patients with diagnosed bladder cancer. The clinical and pathological characteristics of the cohorts are summarized in Table 1 and Appendix A. All the tumors were of urothelial origin. Only samples with more than 60% tumor content were included in the study. The age range was 44–88 with an average age of 72.8. The majority of the patients were male (45/55–81.7%). Nineteen patients (34.55%) in the group suffered from non-invasive bladder cancer in stage Ta, and 18 patients (32.75%) were in stage T1. The remaining 18 patients (32.75%) were diagnosed with invasive bladder cancer in stage T2. Tumor stage was determined according to the 2002 UICC TNM classification, and histological grading was assessed in accordance with the World Health Organization/International Society of Urological Pathology criteria of 2004 [16,17]. A progressive disease was defined as a disease that had progressed to stage T2 or higher, the development of nodal or distant metastases, or death. The control group consisted of 30 patients admitted to the urology ward. They underwent control cystoscopy aimed at confirming or excluding tumor changes in the bladder. The examination did not reveal any tumor changes.

A MirVana™ miRNA Isolation Kit (Life Technologies, Cat No. 1560, Foster City, CA, USA) was used to isolate microRNA from the frozen tumor tissues. The whole procedure was carried out in accordance with the instructions of the producers. Briefly: 1 mL of Lysis/Binding buffer was added to each sample (1 mL per 0.1 g of tissue) and homogenized. After that, 100 µL of miRNA Homogenate Additive was added to sample and incubated for 10 min on ice. Next, 1100 µL of acid-phenol:chloroform was mixed with the sample and centrifuged (5 min at 10,000× *g*). The aqueous phase was transferred to a fresh tube and vortexed with 200 µL of 100% ethanol. A lysate/ethanol mixture was pipetted onto the filter cartridge and centrifuged (15 seconds at 10,000× *g*). The filtrate was collected, and the step was repeated. After that, 400 µL of 100% ethanol was added to the filtrate, pipetted onto new filter cartridge, and centrifuged in the same condition. Two washing steps were conducted: (1) 700 µL of miRNA Wash Solution 1 was applied to the filter cartridge, and (2) 500 µL of Wash Solution 2 and 3 were applied to the filter cartridge; this was repeated twice (at each step, samples were centrifuged for 15 seconds at 10,000× *g*). In the last step, 70 µL of the preheated (95 °C) elution solution was applied to the filter cartridge, which was then spun for 30 seconds at 16,000 g. The collected eluate was stored at −20 °C. An additional DNase and digestion step was performed. The obtained microRNA concentrations were monitored using the spectrophotometric method on the NanoDrop^®^ ND-1000 instrument (NanoDrop Technologies, Wilmington, DE, USA). The purity measurement of the obtained extracts used the relationships A260/230 and A260/280. It is accepted that for good quality nucleic acids, these relationships are, respectively, 1.8–2.2 and 1.8–2.0. The measurement results of the samples selected for further analysis met the required criteria. The purity of the samples was also verified using a Qubit microRNA Assay Kit (Invitrogen, Cat No. Q32880). For reverse transcription, 10 ng of RNA was taken. MiRNAs (hsa-mir-10a, hsa-mir-20a, hsa-mir-21, hsa-mir-130b, hsa-mir-145, hsa-mir-182, hsa-mir-205, and hsa-mir-103) for 55 samples were reverse transcribed using a TaqMan MicroRNA Reverse Transcription Kit (Applied Biosystems Cat No. 4366596) and a 50 nM pool of miRNA specific stem loop primers (Applied Biosystems Cat No. 4427975; details and ID of assays specified in Appendix A) following the manufacturer’s protocol (100 mM dNTPs 0,15 µL, MultiScribe™ Reverse Transcriptase, 50 U/μL 1 µL, 10× Reverse Transcription Buffer 1 µL, RNase Inhibitor, 20 U/μL 0,19 µL, and nuclease-free Water 4.16 µL). The reaction mixtures were incubated at 16 °C for 30 min, at 42 °C for 30 min, and at 85 °C for 5 min (Applied Biosystems microAmp Optical 96-well reaction plate Cat No. N8010560, Micro Amp optical adhesive film Cat No. 4311971), and then the products of the reaction were stored at −20 °C until use. Purity and quantity were verified using a Qubit dsDNA HS assay kit (Invitrogen, Cat No. Q32851).

Real-time polymerase chain reactions (rt-PCR) were performed on CFX96 (BioRad, Hercules, CA, USA) including related documentation with regard to the specific items of MIQE guidelines (Appendix A) [18]. Each sample was run in duplicate at a final volume of 18 µL containing 10 µL of TaqMan 2× Universal PCR Master mix II with no UNG (Applied Biosystems Cat No. 4440040), 7 µL of nuclease free water, and 1 µL of TaqMan^®^ Small RNA Assay (20×). Each PCR included no template control, and all of them were negative. The reaction was heated to 90 °C for 10 min, 55 °C for 2 min, and 72 °C for 2 min, followed by 50 cycles. The mean threshold cycle value (Ct) was used for downstream analyses. miR-103-5p was chosen as an endogenous control. The ΔΔCt method, also defined as the comparative method, was applied in order to mark the expression level of the examined microRNAs [19]. This method is based on mathematical calculations that enable us to determine the relative difference in the expression level of the tested marker between unknown samples and the reference. The first stage consists of the analysis of the marked Ct (the cycle at which the fluorescence level reaches a certain amount/threshold) in the amplification reaction of the examined microRNAs and control microRNA for both the tested and the control groups. The calculated expression level of each patient was normalized against the endogenous control, which was miR-103a-5p [20]. After that, the difference of the tested and control microRNAs (ΔCt) was calculated for individual samples. The calculations were made for both the unknown and control samples.

(ΔCt) (tested group) = Ct miRNA target – Ct miRNA reference

(ΔCt) (control group) = Ct miRNA target – Ct miRNA reference

Next, ΔΔCt was calculated for each sample:

ΔΔCt = ΔCt (tested samples) – ΔCt (median of the control group)

The calculation of the normalized value of the relative expression level (FC) of the tested marker in the tested sample against the control sample was made as follows:(1)$$FC\;=\;2^{-\Delta \Delta Ct}$$

The 2^−ΔΔCt^ method assumes a uniform PCR amplification efficiency of 100% across all samples. In our study, the efficiency was between 98.9% and 100%.

## 3. Data Analysis

The statistical calculations were made using the program STATISTICA 13, Stat-Soft Inc. The differential miRNA expression between bladder cancer cases and controls was determined using Student’s t-statistics. In fact, the distribution of variables differed from the standard normal distribution; therefore, non-parametric tests were applied. The analysis of the unrelated variables was made with the Mann–Whitney U test. The value *p* ˂ 0.05 was accepted as the threshold of statistical difference or correlation significance. Kaplan–Meier analyses with a long-rank test and Cox regression were performed for overall survival time (OS), time to recurrence, and time to progression. The discriminating capacity of miRNAs was assessed by a receiver operating characteristics (ROC) analysis.

## 4. Results

In the first stage of the analysis, the relationship between the abnormal expression of selected microRNAs and other clinical parameters was examined. The raw data from the Ct for individual miRNAs were recalculated for fold change (FCmiR) (Appendix A). In the case of miR-205-5p, all the patients were classified into the reduced expression (low expression—LE) group, while in the case of miR-130b-3p, miR-20a-5p and miR-10a-5p, all the patients were classified into the increased expression (high expression—HE) group. These miRNAs did not differentiate the patients according to clinicopathological parameters; therefore, only FCmiR145, FCmiR21 and FCmiR182 were selected for further analysis. Table 1 presents the probability values (*p*) of the relevant statistics used to make conclusions regarding the existence of relationships between individual variables. The analysis was performed depending on the fulfilled assumptions by the a classic Chi^2^ test, *V*-square test (*V*), or with Yates’s correction (*Y*). There were not any significant correlations observed. Additionally, we did not observe any significant differences for the division of the tested group into Expression 1 (when at least one of the miRNAs indicates abnormal expression) and Expression 2 (when at least two of the analyzed miRNAs show changes). The results are presented in Table 1.

The next step was comparing differences between the level of expression in different stages (TaT1and T2) or grades (LG and HG tumors). The question that was sought next was whether the selected miRNAs could be prognostic classifiers for patients at different stages or grades of cancer. For this purpose, the patients were divided into two groups: 0—patients with stage T2 or higher; 1—patients with stage Ta or T1. The results are presented in Table 2 and Figure 1. The parametric t-test tested the one-sided hypothesis that miRNA for TaT1 < miRNA for T2 and above. The remaining *p*-values were read from the Mann–Whitney U test, which compares distributions (medians). This test is less powerful than the t-test, but it is the only one for non-normal distributions. NMIBC patients (TaT1 in our study) with an LG disease were characterized by significantly higher miR-130b-3p expression values compared to patients with HG tumors. If we consider patients with the LG disease, miR-205-5p, miR-182-5p and miR-20a-5p differentiated this group with BC in stage TaT1 from patients in a higher stage (*p* < 0.05). If we focus on a group of patients in HG, it is miR-130b-3p which best differentiated patients in terms of stage.

To assess the clinical relevance of all miRNAs, a Kaplan–Meier analysis with the log-rank test and Cox regression analyses were performed for overall survival, recurrence-free survival, and progression-free survival (results presented in Table 3 and Figure 2). We did not find any significant differences. Univariate Cox regression was performed to assess the factors that affect the risk of progression, recurrence or death (results presented in Table 4). It has been shown that the older patients are, the higher the risk of death (increasing each year by 30%). In addition, the risk of death for people with stage T2 of the disease is more than six times higher than for patients in stage Ta or T1. People with disease progression are nine times more likely to die. The increase in the expression of miR-205-5p, miR-145-5p and miR-21-5p makes the risk of death higher by 13%, 0.03%, and 0.009%, respectively (Table 4). These percentages result from the interpretation of the hazard ratio (HR) of important risk parameters, for which the Cox regression analysis for OS had *p* < 0.05. The risk of recurrence in patients in stage Ta is over two times higher than people in stage T1 or T2. In the group of patients with recurrence, the death rate is four times lower. The increased expression of miR-20a-5p and miR-182-5p heightens the risk of recurrence by 5% and 6%, respectively. More advanced age increases the risk of progression (by 7% each year). This risk is more than three times higher for people in the T2 stage compared to people in the Ta or T1 stages.

The next step of the analysis was based on results of the area under receiver-operating characteristic curves (ROC). Based on the data from 55 patients with BC and from 30 patients of the control group, an attempt was made to find out which miRNAs, among the selected ones, are the best potential cancer classifier. The conclusion about the significant influence of individual miRNAs on the classification of patients was formed using the multivariable logistic regression model (a logistic regression model with many explanatory variables). The results are presented in Figure 3, Table 5 and Appendix A. The Mann–Whitney U test showed that the distribution of miR-130b-3p was not significantly different for high grade (HG), Ta, and TaT1 patients. Only for the low grade (LG) group did all miRNAs have significantly different distributions compared to the control group. Figure 3 and Figure 4 present the results. Mir-205-5p seems to be a good classifier for LG and HG patients and also for Ta and T1 stages. Logistic regression assessed with a backward elimination approach resulted in a pattern of three miRNAs (miR-205-5p, miR-20a-5p and miR-182-5p). For HG, the relevant classifiers are miR-205-5p and miR-20a-5p, which gave an AUC = 0.964, whereas low LG miR-205-5p and miR-182-5p gave an AUC = 0.992. The model classifies HG as well as BC. The results are presented in Figure 4 and Table 6.

## 5. Discussion

The progression in bladder cancer is a complex and multifactorial process [21,22]. In oncology, histopathological examination is still the most important method to determine the diagnosis and classification of tumors; however, current prognosticators such as the tumor grade, stage, size, and multifocality do not accurately reflect clinical outcomes and have limited usefulness for a reliable risk-adjusted therapy decision. At present, there are not enough good markers that could be used as tools to support screening, detecting or monitoring the disease [23,24]. miRNA is an “attractive candidate” as a potential diagnostic and prognostic biomarker, not only due to its high level of stability in body tissues and fluids but also due to its ability to be quantified in relatively easy and cheap techniques like real-time PCR [25,26]. Various miRNAs have been identified as important targets in bladder cancer development, but the large number of different expression profiling platforms such as microarrays, miRCURY ready to use PCR, TaqMan Human MicroRNA Probes, and different reference genes used for normalization are the reason that the results are not comparable and it is difficult to put miRNAs into clinical practice. Therefore, obtaining reliable, not biased miRNA expression data is crucial for selecting clinically useful markers.

It is estimated that over 30% of the protein-coding genes in human cells are controlled by miRNAs. One type of miRNA can even control the expression of hundreds of target genes, and one gene can be controlled by numerous miRNAs. These molecules are regarded as the “key” ones in the gene regulatory network. MiRNAs are involved in many significant biological processes, such as apoptosis, proliferation, cell diversification, and oncogenesis. In this study, we compared the expression of selected miRNAs in non-malignant and malignant bladder tissue, and we identified three down-regulated ones (miR-205-5p, miR-182-5p, and mir-145-5p) and two up-regulated (miR-20a-5p and miR-130b-3p). In previous studies, all of them have been found to be differently expressed in malignant bladder tissue (mainly the underexpression of miR-145-5p and the overexpression of the others), but in this study, the normalization of expression data was performed using miR-103-5p as a reference [25,26]. An endogenous control, in relation to which we normalized the results of other miRNAs, should, as a rule, show stability in a given tissue. In reality, this is very difficult to achieve, and different groups of researchers choose different controls and obtain different results due to these controls. Ratert at al. confirmed that using RNU6B and RNU48 could lead to seriously biased results regarding miRNA expression analysis [27]. Peltier and Latham found that some miRNAs (including miR-106a and miR-191) were the most consistently expressed across different human tissues [28]. They also observed that RNU6 and RNA5S were the least stable. Hofbauer et al. used two endogenous controls in their research, RNU48 and miR-103-5p, and they achieved satisfactory results [29]. In our research, we followed the results of others, including the possibility of the use of miR-103-5p as an endogenous control in commercial sets (Exiqon, Vedbaek Denmark). The studies of Boisen et al. and Parvaee showed that mir-103-5p expression assessed in isolation from formalin-fixed parafin-embedded (FFPE) cancer tissue was the most stable reference miRNA in colorectal (CRC), pancreatic (PC), and intestinal type gastric cancer [30,31].

We subdivided the tumor samples in terms of the low and high grade diseases. The comparison of the miRNAs revealed four significant differentially expressed miRNAs (miR-205-5p, miR-130b-3p, miR-20a-5p, and miR-182-5p). Several studies have implicated miRNAs as prognostic markers for BC. As already shown in previous studies, miR-205-5p expression in normal and tumor samples seems to be coordinated with the *mir-8* family. Lenherr found abnormal expression between progressors and non-progressors for several miRNAs including miR-205-5p and miR-20a-5p. Some of the known targets of miR-205-5p include *ZEB1/2*, *PTEN*, and *VEGFA* [32]. The downregulation of miR-205-5p has been linked to the epithelial-mesenchymal transition (EMT) and has been significantly associated with progression in non-muscle invasive BC. However, the results obtained by different research groups are not consistent due to factors that were already-mentioned in the introduction (differences in the chosen methods). Contrary to that, Dip et al. observed that miR-205-5p was overexpressed in pT2–3 stages of BC [33]. In their study, miR-10a-5p overexpression was associated with shorter disease-free and disease specific survival. Ecke et al. did not confirm the statistical significance for differences in the expression of miRNA-205-5p between non-malignant and BC samples, but they detected a statistically significant reduction in the expression of miR-130b-3p (the best discriminator, also shown in our research) [34]. miR-145-5p overexpression inhibited cell proliferation and migration in BC [35]. Moreover, the downregulation of mir-145-5p was found to be directly targeting the *TAGLN2* gene (its increased expression promoted cell proliferation and migration). Li et al. also confirmed the correlation between the overexpression of miR-145-5p and poor survival [36]. Unfortunately, we failed to achieve such correlation. Inamoto et al. also confirmed the deregulation of miR-145-5p expression and its association with the aggressive phenotype, but they showed its protective effect. miR-145-5p expression was significantly lower in BC samples and cell lines compared to those in normal bladder tissue [37]. Pignot et al. observed that most of the examined miRNAs were deregulated in the same way in the two types of bladder cancer, irrespective of the pathological stage [38]. In their study, miR-182-5p was downregulated and was found to be related to tumor aggressiveness (associated with both recurrence-free and overall survival in univariate analysis). In our study, the high expression of mir-182-5p and mir-20a-5p correlated with the risk of disease recurrence (Table 4; risk higher by 0.06% and 0.0002%, respectively). Urquidi et al. identified a few miRNA set classifiers for predicting the presence of bladder cancer (25 miRNAs, 20 miRNAs, 15 miRNAs, and 10 miRNAs), but none of them included those ones which we found in our study [39]. The authors note that these biomarkers were correlated with the presence of BC, but their association with clinical variables was much less evident. In our opinion, different sets of miRNAs can be suggested as prognostic biomarkers (three: miR-9, mir-183, and mir-200b; two: miR-143 and miR-145); however, until now, only one study had verified the examined miRNAs as independent markers [27]. Ecke et al. identified miR-199a-3p and miR-214-3p as independent prognostic biomarkers for the prediction of overall survival (OS) in MIBC patients after radical cystectomy (RC). They used a combination of four miRNAs (miR-101, miR-125a, miR-148b, and miR-151-5p) or three miRNAs (miR-148b, miR-181b, and miR-874) as endogenous controls. The study was carried out in a formalin-fixed, paraffin-embedded (FFPE) tissue specimen. These markers were not evaluated by us. Ecke at al. also analyzed the expression of miR-205-5a, but they did not confirm its usefulness. It needs to be stressed, however, that the marking was done in FFPE, whereas our tests were carried out in fresh, frozen tumor tissue. Armstrong’s results for matched tumor and bio-fluids in BC showed that there is an overlap between the expression of miRNAs in different bio-specimen sources, but overexpression in all three kinds of the biological samples has only been observed for two tested miRNAs (miR-4454 and miR-21) [40]. No correlation has been observed between expression in tumors and plasma exosomes (using the NanoString nCounter microRNA assay technique). In their review, Lee et al. showed a correlation in the changes of the expression of miRNAs isolated from bladder cancer tissues and urine (in multiple results) for only 14 miRNAs, including miR-145, miR-182, and miR-205 [15]. On the other hand, Baumgart et al. observed that nine miRNAs were consistently differently expressed in both invasive cells and their secreted exosomes, but the remaining six miRNAs were only dysregulated in exosomes [41]. The NanoString technique has its advantages, as it does not require the application of any nucleoid acids. However, it is expensive and hardly available. A real-time PCR technique is available, but any obtained result is affected by many factors, such as the kind of tissue, the way of normalization, and the way of analysis.

Receiver operating characteristics analyses showed a good ability to discriminate between non-malignant and malignant tissues for the investigated miRNAs. Based on binary logistic regression using the backward elimination approach, the optimal combination for discriminating healthy people from BC patients is miR-205-5p, miR-20a-5p and miR-182-5p (AUC > 0.9; *p* < 0.05). Lv et al. Egawa et al. and Liu et al. also confirmed that miR-130b-3p could play a critical role in the development and progression of bladder cancer [42,43,44]. Fang et al. found an miR-205-5p area under the receiver-operating characteristics curve value of 0.950 for discriminating BC patients from healthy people and a value of 0.668 for discriminating MIBC from NMIBC [10]. The log-rank test and univariate and multivariate Cox regression analyses did not indicate that high miR-205-5p expression in NMIBC patients was associated with cancer specific survival.

We faced some limitations in our study, one of which was a relatively small group of patients. We only tried to use the samples that were characterized by adequate amounts of tumor cells. It is not easy to obtain a large group of patients with bladder cancer progression who can provide biological material for tests, as such cases constitute the minority in this disease. The applied study technique is relatively cheap and easy. Thus, it could be used for examining chosen markers on a daily basis.

## 6. Conclusions

This study follows the strategy “from top to bottom,” which means choosing the phenotype of patients (histopathological characteristics and survival) and evaluating the molecular markers of such a phenotype. The goal for the future is the opposite course of analysis, which is “from genotype to phenotype.” Based on the detection of diagnostically and prognostically significant differences between normal and cancer samples, we could assess the biological potential of the tumor and its aggressiveness. As a result, we could enable the choice of appropriate therapeutic measures, tailored to individual patients; this is personalized medicine. Finally, we could lengthen a patient’s life and improve its quality without offering a radical treatment if it is not necessary. The implementation of miR-205-5p, miR-20a-5p and miR-130b-3p into routine practice can be an alternative to screening or the follow up of treatment effects. Such analyses can help in the search of non-invasive markers, especially since they can also be evaluated in urine or plasma. Our findings could be of clinical importance, but the results should be validated in a bigger group.

## Figures and Tables

**Figure 1 cancers-11-01551-f001:**
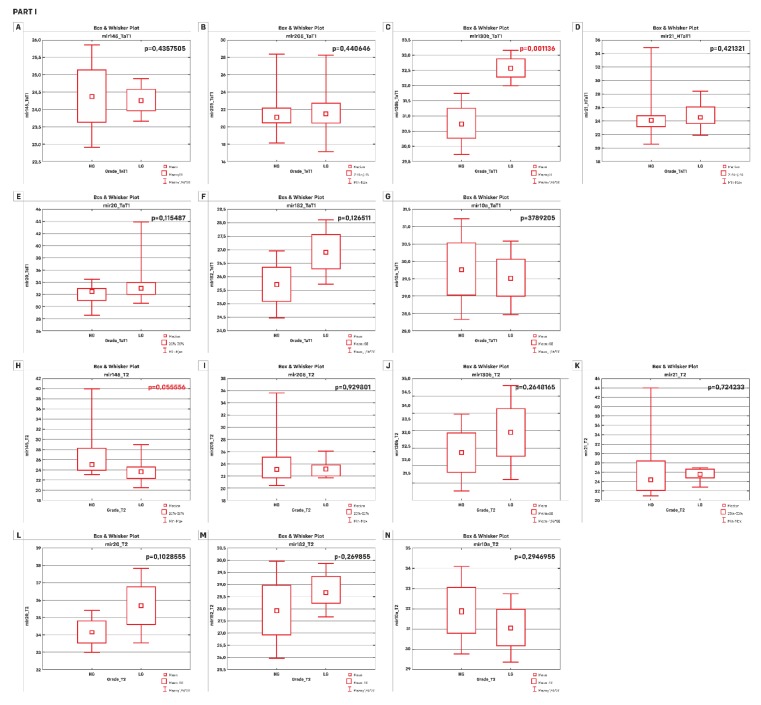
Differences in expression level for patients: Group 0—BC patients in T2; Group 1—BC patients in stage TaT1 or in low or high grade group (low grade (LG) or high grade (HG)). Part I presents results for the differentiation of patients in terms of grade in TaT1 (**A**–**G**) and T2 (**H**–**N**) groups. Only miR-130b-3p differentiated patients in stage TaT1 according to grade (miR-145-5p was close to significance). Part II presents results for the differentiation of patients in terms of stage in high (**A**–**G**) or low grade (**H**–**N**) groups. If we take patients with LG, the differentiating miRNAs in terms of stage were miR-205-5p, miR-20a-5p and miR-182-5p. In the case of patients in the HG group, miR-20a-5p, miR-205-5p and miR-182-5p always had significantly lower values of expression in patients in the TaT1 stage of the disease compared to patients in the T2 stage. (**A**) and (**H**) miR-145-5p; (**B**) and (**I**) miR-205-5p; (**C**) and (**J**) miR-130b-3p; (**D**) and (**K**) miR-21-5p; (**E**) and (**L**) miR-20a-5p; (**F**) and (**M**) miR-182-5p; and (**G**) and (**N**) miR-10a-5p. *p*-values in read are significant (*p* < 0.05).

**Figure 2 cancers-11-01551-f002:**
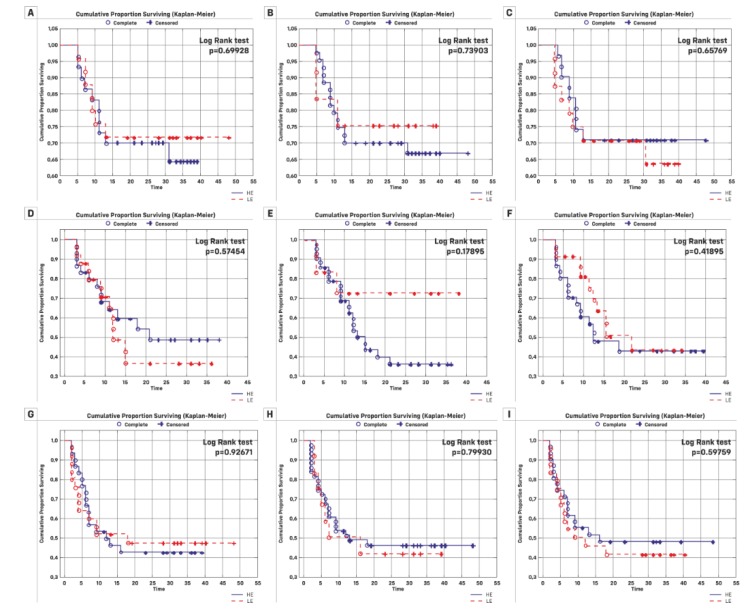
Kaplan–Meier plots for patients’ group divided into low expression (LE) and high expression (HE): Cancer-specific survival (**A**) miR-145-5p, (**B**) miR-21-5p, and (**C**) miR-182-5p; recurrence free-survival (**D**) miR-145-5p, (**E**) miR-21-5p, and (**F**) miR-182-5p; and progression-free survival (**G**) miR-145-5p, (**H**) miR-21-5p, and (**I**) mir-182-5p.

**Figure 3 cancers-11-01551-f003:**
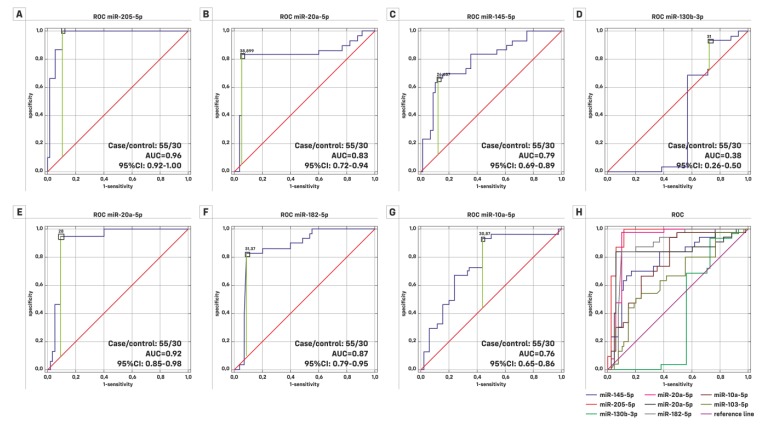
The receiver operating characteristics (ROC) curves for BC prediction using the expression level of (**A**) miR-205-5p, (**B**) miR-20a-5p, (**C**) miR-145-5p, (**D**) miR-130b-3p, (**E**) miR-21-5p, (**F**) miR-182-5p, (**G**) miR-10a-5p and (**H**) summary of curves for all miRNAs. The best classifiers are miR-205-5p, miR-20a-5p and miR-145-5p, as these could significantly discriminate BC patients from the control group by an AUC higher than 0.9; *p* < 0.05.

**Figure 4 cancers-11-01551-f004:**
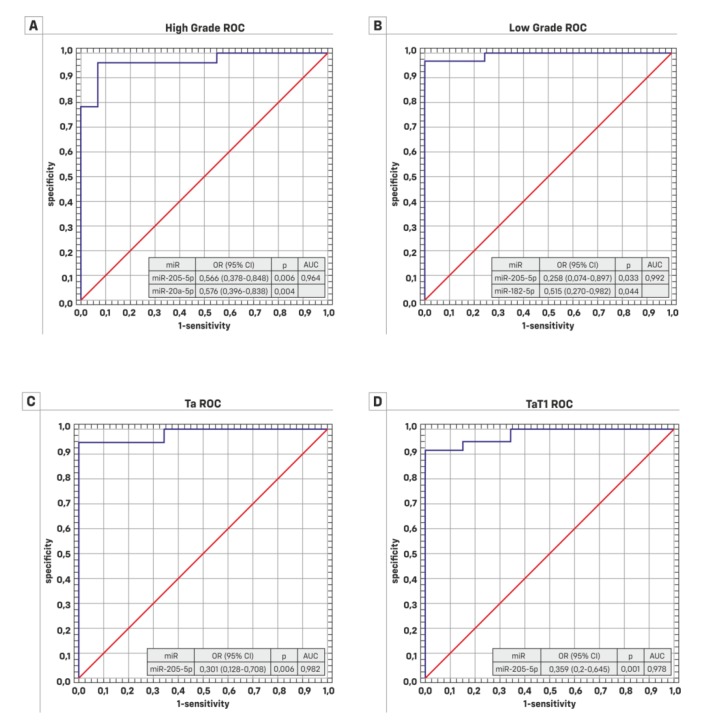
Multivariate logistic model of one or two signature microRNAs (miRNAs): (**A**) For high grade tumors, miR-205-5p and miR-20a-5p are the best classifiers; AUC = 0.964. (**B**) For low grade tumors, miR-205-5p and miR-182-5p are the best classifiers; AUC = 0.992. (**C**) for Ta and (**D**) for TaT1, mir-205-5p is the best classifier; AUC = 0.982 and AUC = 0.978 respectively. (OR—odds ratio; CI—confidence interval; AUC—area under curve).

**Table 1 cancers-11-01551-t001:** Differences in expression level of selected miRNAs in the patients’ group according to clinicopathological parameters.

		*FCmiR-145*		*p*-Value	*FCmiR-21*		*p*-Value	*FCmiR-182*		*p*-Value	Abnormal Expression 1		*p*-Value	Abnormal Expression 2		*p*-Value
Clinicopathological Parameters		HE n (%)	LE n (%)		HE n (%)	LE n (%)		HE n (%)	LE n (%)		Yes n (%)	No n (%)		Yes n (%)	No n (%)	
Total	55															
Sex																
	Female	4 (7.27%)	6 (10.91%)		3 (5.45%)	7 (12.73%)		5 (9.09%)	5 (9.09%)		7 (12.73%)	3 (5.45%)		3 (5.45%)	7 (12.73%)	
	Male	26 (47.27%)	19 (34.55%)	0.503 (Y)	9 (16.36%)	36 (65.45%)	0.787 (Y)	26 (47.27%)	19 (34.55%)	0.923 (Y)	34 (61.82%)	11 (20%)	0.971 (Y)	20 (36.36%)	25 (45.45%)	0.629 (Y)
Age at Diagnosis																
	<60	2 (3.64%)	4 (7.27%)		1 (1.82%)	5 (9.09%)		6 (10.91%)	0 (0%)		6 (10.91%)	0 (0%)		2 (3.64%)	4 (7.27%)	
	>60	28 (50.91%)	21 (38.18%)	0.502 (Y)	11 (20%)	38 (69.09%)	0.841 (Y)	25 (45.45%)	24 (43.64%)	0.064 (Y)	35 (63.64%)	14 (25.45%)	0.308 (Y)	21 (38.18%)	28 (50.91%)	0.994 (Y)
Smoking Status																
	Yes	23 (41.82%)	23 (41.82%)		9 (16.36%)	37 (67.27%)		26 (47.27%)	20 (36.36%)		34 (61.82%)	12 (21.82%)		17 (30.91%)	29 (52.73%)	
	No	7 (12.73%)	2 (3.64)	0.244 (Y)	3 (5.45%)	6 (10.91%)	0.636 (Y)	5 (9.09%)	4 (7.27%)	0.753 (Y)	7 (12.73%)	2 (3.64%)	0.861 (Y)	6 (10.91%)	3 (5.45%)	0.199 (Y)
Occupatinal Exposure																
	Yes	21 (38.18%)	19 (34.55%)		6 (10,91%)	34 (61.82%)		21 (38.18%)	19 (34.55%)		28 (50.91%)	12 (21.82%)		15 (27.27%)	25 (45.45%)	
	No	9 (16.36%)	6 (10.91%)	0.622 (V)	6 (10.91%)	9 (16.36%)	0.102 (Y)	10 (18.18%)	5 (9.09%)	0.349	13 (23.64%)	2 (3.64%)	0.359 (Y)	8 (14.55%)	7 (12.73%)	0.293 (V)
Tumour Stage																
	Ta	9 (16.36%)	10 (18.18%)		1 (1.82%)	18 (32.73%)		11 (20%)	8 (14.55%)		14 (25.45%)	5 (9.09%)		6 (10.91%)	13 (23.64%)	
	T1	10 (18.18%)	8 (14.55%)		6 (10.91%)	12 (21.82%)		9 (16.36%)	9 (16.36%)		13 (23.64%)	5 (9.09%)		8 (14.55%)	10 (18.18%)	
	T2	11 (20%)	7 (12.73%)	0.699	5 (9.09%)	13 (23.64%)	0.089	11 (20%)	7 (12.73%)	0.786	14 (25.45%)	4 (7.27%)	0.924	9 (16.36%)	9 (16.36%)	0.505
Grade																
	high grade	13 (23.64%)	9 (16.36%)		5 (9.09%)	17 (30.91%)		12 (21.82%)	10 (18.18%)		16 (29.09%)	6 (10.91%)		11 (20%)	11 (20%)	
	low grade	17 (30.91%)	16 (29.09)	0.580	7 (12.73%)	26 (47.27%)	0.841 (Y)	19 (34.55%)	14 (25.45%)	0.826 (V)	25 (45.45%)	8 (14.55%)	0.802 (V)	12 (21.82%)	21 (38,18%)	0.319 (V)
Recurrence																
	Yes	13 (23.64%)	13 (23.64%)		3 (5.45%)	23 (41.82%)		16 (29.09%)	10 (18.18%)		21 (38.18%)	5 (9.09%)		9 (16.36%)	17 (30.91%)	
	No	17 (30.91%)	12 (21.82%)	0.521	9 (16.36%)	20 (36.36%)	0.083 (V)	15 (27.27%)	14 (25.45%)	0.463	20 (36.36%)	9 (16.36%)	0.320 (V)	14 (25.45%)	15 (27.27%)	0.305
Progression																
	Yes	17 (30.91%)	13 (23.64%)		7 (12.73%)	23 (41.82%)		16 (29.09%)	14 (25.45%)		21 (38.18%)	9 (16.36%)		14 (25.45%)	16 (29.09%)	
	No	13 (23.64%)	12 (21.82%)	0.729	5 (9.09%)	20 (36.36%)	0.767 (V)	15 (27.27%)	10 (18.18%)	0.619	20 (36.36%)	5 (9.09%)	0.401 (V)	9 (16.36%)	16 (29.09%)	0.424
Death																
	Yes	10 (18.18%)	7 (12.73%)		3 (5.45%)	14 (25.45%)		9 (16.36%)	8 (14.55%)		11 (20%)	6 (10.91%)		9 (16.36%)	8 (14.55%)	
	No	20 (36.36%)	18 (32.73%)	0.673 (V)	9 (16.36%)	29 (52.73%)	0.882 (Y)	22 (40%)	16 (29.09%)	0.734	30 (54.55%)	8 (14.55%)	0.432 (Y)	14 (25.45%)	24 (43.64%)	0.267 (V)

Y-test chi-squared with Yeats corrections; V-test V-squared.

**Table 2 cancers-11-01551-t002:** Differences in expression in A) stage TaT1 and T2 according to different grade and B) low and higf grade tumors according to stage of BC.

A)	TaT1 *p*-value	T2 *p*-value
miR-145-5p	0.4357505 *	0.055556
miR-205-5p	0.440646	0.929801
miR-130b-3p	**0.001136 ***	0.2648165 *
miR-21-5p	0.421321	0.724233
miR-20a-5p	0.115487	0.1028555 *
miR-182-5p	0.126511 *	0.269855 *
miR-10a-5p	0.3987205 *	0.2946955 *
B)	HG *p*-value	LG *p*-value
miR-145-5p	0.132994	0.336568 *
miR-205-5p	0.065169	**0.030956 ***
miR-130b-3p	**0.00531 ***	0.138824 *
miR-21-5p	0.606318	0.141797 *
miR-20a-5p	**0.019231**	**0.038561**
miR-182-5p	**0.037793 ***	**0.015572 ***
miR-10a-5p	0.06102 *	0.081524 *

Results obtained with Mann-Whitney U test (*p*-values without *) and parametric *t* test (*p*-values with *). Bold face represents *p*-value <0.05.

**Table 3 cancers-11-01551-t003:** Kaplan-Meier analysis for overall survival, time to recurrence and time to progression in the patients’ group.

		Kaplan-Meier Analysis					
		Overall Survival		Recurrence		Progression	
	Overall n (%)	Rate	Log-Rank Value	Rate	Log-Rank Value	Rate	Log-Rank Value
Total	55						
***FCmiR-145***							
HE	30	10		13		17	
LE	25	7	0.6992	13	0.5745	13	0.9267
***FCmiR-21***							
HE	12	3		3		7	
LE	43	14	0.7390	23	0.1789	7	0.7993
***FCmiR-182***							
HE	31	9		16		16	
LE	24	8	0.6576	10	0.4189	14	0.5976
Total	55						
**Abnormal expression 1**							
Yes	41	11		21		21	
No	14	6	0.2875	5	0.3499	9	0.2847
**Abnormal expression 2**							
Yes	23	9		9		14	
No	32	8	0.2551	17	0.6881	16	0.5205

LE—low expression; HE—high expression.

**Table 4 cancers-11-01551-t004:** Univaraite Cox regression analysis of potential predictor variables and overall survival, time to recurrence and time to progression in the group of patients (n = 55).

		Overall Survival				Time to Recurrence				Time to Progression		
	Beta	HR (95% CI)	*p*-value	*p*-value for Chi^2^	Beta	HR (95% CI)	*p*-value	*p*-value for Chi^2^	Beta	HR (95% CI)	*p*-value	*p*-value for Chi^2^
Gender	−0.57	0.56 (0.13–2.47)	0.448	0.415	−0.816	0.44 (0.13–1.47)	0.184	0.142	0.029	1.03 (0.42–2.42)	0.948	0.948
Age at diagnosis	0.266	1.30 (1.17-1.45)	**0.000**	0.000	-0.002	0.997 (0.99-1.005)	0.526	0.524	0.068	1.07 (1.02-1.12)	**0.0034**	0.003
Stage												
Ta–T1&T2	−0.013	0.98 (0.36–2.67)	0.97	0.98	0.738	2.09 (0.97–4.53)	**0.0607**	0.061	−1.388	0.25 (0.09–0.65)	**0.005**	0.0013
Ta&T1–T2	1.821	6.17 (2.25–16.89)	**0.0004**	0.00028	−0.766	0.46 (0.16–1.35)	0.159	0.126	1.109	3.03 (1.46–6.26)	**0.0027**	0.0034
Occupatinal Exposure	1.12	3.08 (0.7–13.47)	0.135	0.087	−0.669	0.51 (0.23–1.13)	0.097	0.108	0.874	2.39 (0.91–6.28)	0.075	0.052
Grade	2.85	17.36 (3.89–77.41)	**0.00018**	0.000	−1.615	0.19 (0.06–0.66)	**0.008**	0.001	1.775	5.89 (2.59–13.38)	**0.00002**	0.00001
Smoking Status	0.37	1.45 (0.33–6.37)	0.619	0.603	0.101	1.11 (0.38–3.21)	0.852	0.85	−0.07	0.93 (0.36–2.43)	0.885	0.886
Recurrence	−1.28	0.28 (0.09–0.86)	**0.026**	0.015					−2.229	0.107 (0.04–0.28)	**0.000008**	0.00000
Progression	2.16	8.67 (1.97–8.13)	**0.004**	0.00031	−1.717	0.18 (0.07–0.48)	**0.0006**	0.00008				
FCmiR-145	0.0003	1.0003 (1.00009–1.0006)	**0.0069**	0.038	−0.019	0.98 (0.93–1.03)	0.393	0.099	0.0001	1.0001 (0.99–1.0003)	0.243	0.321
FCmiR-205	0.12	1.13 (1.03–1.24)	**0.0089**	0.045	−0.167	0.85 (0.36–1.96)	0.697	0.521	0.046	1.05 (0.97–1.13)	0.233	0.311
FCmiR-130b	0.0003	0.99 (0.99–1.00)	0.466	0.398	0.0003	1.0003 (0.99–1.0007)	0.131	0.176	−0.0002	0.99 (0.99–1.00)	0.484	0.437
FCmiR-21	0.00009	1.00009 (1.000025–1.00015)	**0.0069**	0.038	0.0004	1.0000006 (0.98–1.006)	0.145	0.156	0.00003	1.00003 (0.99–1.00008)	0.259	0.336
FCmiR-20a	−0.00013	0.999 (0.999–1.0)	0.412	0.177	0.000002	1.000002 (1.0–1.000003)	**0.031**	0.097	−0.00013	0.999 (0.999–1.0)	0.412	0.177
FCmiR-182	−0.034	0.966 (0.87–1.07)	0.529	0.172	0.0006	1.0006 (0.00004–1.001)	**0.035**	0.104	−0.0009	0.999 (0.995–1.002)	0.599	0.243
FCmiR-10a	−0.0004	0.999 (0.997–1.001)	0.672	0.47	−0.0004	0.999 (0.998–1.0007)	0.505	0.301	0.0003	1.0003 (0.999–1.0006)	0.129	0.218
Abnormal Expression 1	−0.5328	0.587 (0.217–1.588)	0.294	0.309	0.4376	1.549 (0.583–4.11)	0.379	0.358	−0.4315	0.649 (0.297–1.42)	0.279	0.295
Abnormal Expression 2	0.5419	1.719 (0.663–4.459)	0.265	0.265	−0.1626	0.85 (0.378–1.91)	0.694	0.691	0.2274	1.255 (0.612–2.575)	0.535	0.536

Bold face representing *p*-values <0.05; CI—Coincidence Interval; HR—Hazard Ratio.

**Table 5 cancers-11-01551-t005:** Results of Mann-Withney U test in bladder cancer (BC) group and in subgrups divided according to grade or stage.

Mann Whitney U Test	BC Group		Subgroups		
	***p*** **-value**	**HG *p*-value**	**LG *p*-value**	**Ta *p*-value**	**TaT1 *p*-value**
miR-145-5p	**0.000005**	**0.003612**	**0.000002**	**0.000026**	**0.000001**
miR-205-5p	**0.000000**	**0.00000**	**0.00000**	**0.000000**	**0.000000**
miR-130b-3p	0.073733	0.770102	**0.011493**	0.257699	0.479923
miR-21-5p	**0.000000**	**0.000004**	**0.000024**	**0.000000**	**0.000000**
miR-20-5p	**0.000000**	**0.000001**	**0.000001**	**0.000003**	**0.000001**
miR-182-5p	**0.000000**	**0.000009**	**0.00000**	**0.000001**	**0.000000**
miR-10a-5p	**0.000048**	**0.014889**	**0.000016**	**0.000009**	**0.000004**

Bold face represents *p* value <0.05; HG—high grade; LG—low grade.

**Table 6 cancers-11-01551-t006:** ROC characteristics for subgroups of patients with BC (HG-high grade, LG-low grade, Ta stage, TaT1 stages) and control group.

	HG (Case/Control = 22/30)			LG (Case/Control = 33/30)		
ROC Characteristics	AUC	95% Cl	Significance *p*	AUC	95% Cl	Significance *p*
miR-145-5p	0.732	0.591–0.873	**0.0013**	0.833	0.731–0.936	**0.0001**
miR-205-5p	0.941	0.860–1.000	**0.0001**	0.981	0.955–1.000	**0.0001**
miR-130b-3p	0.475	0.287–0.663	0.7964	0.313	0.167–0.458	**0.0115**
miR-21-5p	0.851	0.717–0.984	**0.0001**	0.936	0.866–1.000	**0.0001**
miR-20a-5p	0.87	0.761–0.978	**0.0001**	0.801	0.675–0.927	**0.0001**
miR-182-5p	0.841	0.703–0.976	**0.0001**	0.895	0.809–0.980	**0.0001**
miR-10a-5p	0.696	0.545–0.846	**0.0109**	0.807	0.698–0.916	**0.0001**
		Ta (case/control = 19/30)			TaT1 (case/control = 37/30)	
ROC Characteristics	AUC	95% Cl	Significance *p*	AUC	95% Cl	Significance *p*
miR-145-5p	0.842	0.734–0.950	**0.0001**	0.83	0.728–0.932	**0.0001**
miR-205-5p	0.982	0.947–1.000	**0.0001**	0.978	0.950–1.000	**0.0001**
miR-130b-3p	0.401	0.205–0.597	0.3236	0.448	0.300–0.596	0.493
miR-21-5p	0.939	0.837–1.000	**0.0001**	0.925	0.849–1.000	**0.0001**
miR-20a-5p	0.872	0.764–0.980	**0.0001**	0.83	0.713–0.946	**0.0001**
miR-182-5p	0.888	0.777–0.998	**0.0001**	0.902	0.821–0.983	**0.0001**
miR-10a-5p	0.858	0.738–0.978	**0.0001**	0.817	0.714–0.920	**0.0001**

AUC—Area Under Curve; CI—Coincidence Interval; Bold face representing *p*-values <0.05; ROC—Receiver-operating Characteristcs.

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
