# Peer review of "MicroRNAs Which Can Prognosticate Aggressiveness of Bladder Cancer"

_cancers, 2019, doi:10.3390/cancers11101551_

Round 1

Reviewer 1 Report

The authors addressed all the raised comments. The manuscript can be suitable for publication

Author Response

Thank you very much.

Reviewer 2 Report

The authors have overall addressed my previous comments. Some typos need to be corrected. For example, line 305"control". Figure 1 fond size needs to be enlarged.

Author Response

Thank you very much for comments.

We have corrected typing errors. They are marked in red.

We cannot increase the font size in Figure 1, because these are the original charts from the data analysis program. It is also difficult to obtain because the size of individual graphs in Figure 1 has decreased.This is a consequence of the fact that there are now more graphs. We have increased the font size of the  p value.

Reviewer 3 Report

Line 163: dCT (tested samples) - dCT (control) : Not clear, does this mean that patients were appariated?

or each dCt for the tested sample is compared to the mean (or mediane) of the control group?

Line 391: please remove this sentence from the conclusion: "we have also noted a lot of..." this type of difficulty is largely commented everywhere.

Author Response

Thank you very much for comments.

Point 1:

Line 163 was corrected - it should be "mediane of control group"

Point 2:

Line 391: please remove this sentence from the conclusion: "we have also noted a lot of..." this type of difficulty is largely commented everywhere.

Line 391 - the sentence was removed

Typo corrections are marked in red.

This manuscript is a resubmission of an earlier submission. The following is a list of the peer review reports and author responses from that submission.

Round 1

Reviewer 1 Report

In this paper « microRNA which can prognosticate aggressiveness of bladder cancer », Borkowska EM et al. report that some miRNAs correlate with the bladder cancer progression and would be potential markers to predict bladder cancer and to differentiate low grade from high grade tumors. The question is of interest and the authors explain in the introduction the necessity of these predictions to adapt the patient treatment.

The expression level of 7 pre-selected miRNAs has been measured in frozen tumor tissues by TaqMan RT-qPCR (Applied Biosystems). The differential miRNA expression has been compared between patients in different stages and also according to grade. Moreover it seems that a high expression level of miR-205-5p, -145-5p, -21-5p increases significantly the risk of death. The conclusion is that miR-205-5p and miR-20-5p might be used as classifier for high grade bladder cancer and miR-205-5p and miR-182-5p for low grade.

The main problem of the study is that it is difficult to understand the data because the result section is not well written. For example, line 139, I could not figure out what has been done and the interpretation of the data. These data should be presented in a supplemental table, with the fold changes and patient classification. Then the authors mention that miR-145, -21, -182 were further studied. Does this mean that these 3 miRNAs discriminate the patients? From line 154 to 156-: these 2 sentences have to be re-written properly. Table2 (Tabele2), Change the title to explain what are the data (not given the test used for  title!). I do not understand what the asterisks are. What is the remaining p? miR130b seems significant in A but it is written significant for high grade.Pleas explain properly. Line176: 13%, 0.03%... What these come from??

Please explain the pre-selection of the miRNAs: from literature?? From the discussion (ref 27), the authors mention miR-199a-3p and miR-214-3 to improve prediction: why the authors have not selected these miRNAs? Especially, because the authors explain in the discussion the difficulty to compare results between the labs (platform, technology….). Line101, the miRNAs have to be properly written, with full name, in a sup table, with the miRNA name assay and referencein Applied Biosystems. Has- and not has, miR103-5p, mir145….please standardize the nomenclature. All over the text: sometimes microRNA, miR, microR (line124)….

Method: please fully describe the extraction from the frozen tissue. Line 104: indicate the miR primer concentration for reverse transcription. I guess it is equal concentration for each miRNAs. Amount of RNA used fo the reverse transcription?;  line 115: amount of cDNA (as dilution of the RT sample) in the assay? Why miR-103-5p has been chosen as endogenous control?

Line 119: change the title. The ΔΔCt method is awkwardly presented.  Ct values  recorded  by  the  software ….were  normalized  by  the  Ct  of the endogenous  miRNAs (miR-103a-5p) for  the  relative  quantification  (RQ)  of  miRNA  levels  as  Fold-Change  (FC) = Log2 (2–ΔΔCT). I understand the normalization for ΔCt but what is the calculation for ΔΔCT?

The table legends have to be written properly.

I do not have any supplemental table: line 113.

The abstract should present clear results and define the abbreviations ..NMIBC…

Reviewer 2 Report

In this study, the authors aim to investigate the prognostic value of a few microRNAs for the aggressiveness of bladder cancer. They quantified miRNAs from 55 bladder cancer patients and 30 persons as control. Their Kaplan-Meier, Receiver-operating characteristics (ROC) curves and the area under the curve (AUC) analysis nominate some miRNAs could be used for screening. However, this is a very limited study, and there is inconsistency and some major flaws in the study. The manuscript is confusing to read.

Detailed comments:

1) In the methods, it was indicated that “The purity of the samples was verifiedusing Qubit ssDNA HS assay kit”. However, there is specific kit for measuring microRNA, why authors used ssDNA kit.

2)It is not clear how they authors come up with those 7 candidate miRNAs, what is the rationale, and what are the known functions of them in bladder cancer.

3) There is not enough clarification for choosing miR-103a-5p as endogenous control.

4) All selected miRNAs expression should be shown in table 1.

5) Figure1, the p value should be provided. It is unclear the difference of A and D, the x and y-axis title is confusing. And all miRNAs should be included.

6) The names of the miRNAs should be consistent throughout the paper.

7) Although the authors mentioned that some of the miRNA can be detected in the urine or plasma, the samples used in this study all come from the tissue biopsy, and it is not clear whether these miRNA show similar expression pattern as demonstrated by the authors.

Reviewer 3 Report

The work is interesting, but needs to be improved in order to be published in Cancers.

Listed below some coments:

In the introduction section, the sentence in lines 42-47 is not clear. Authros should rephrase and expand the sentence. Line 47, expand the acronimous MIBC, as the authors do for NMIBC Line 56,57 the sentence is not correct: what does it mean “genes coded fibroblast growth factor receptor “3? Remove the bold font from material and methods section In the first paragraph of the result section, it is not clear how the authors analyze abnormally expressed miRNAs in patients. Explain better. Where did the authors find the miRNAs listed in table 1 and 2? Correct “tabele” with “TABLE”. Explain better table 1 and 2 in the relative legend Enlarge the font size of the figures Why the patients were discriminated into Low expression group while in the following analyses patients with increased expressino of miR205 have an higher increase of death? And how can the autyhors explain why the incresed recurrence reduces the risk of death, and the risk of recurrence is higher in patients in stage Ta compared to stage T1 or T2? In general, authors should put the microRNA in their functional and biological context instead of simply listing their statistical results For example, authors can analyze the expression of some transcripts targeted by the analyzed miRNAs (they should be anticorrelated to these latters) in patients respect to healty controls, in tumoral versus non-tumoral tissue in the same patients, and also their prognostic value.

Mir 103-5p is often modulated in cancer. Why the authors used it for normalization? Provide raw data showing no changes in miR 103-5p abundance in bladder cancer cell lines, in tumoral respect to non tumoral tissues in the same patients, and in bladder cancer patients respect to healty controls. In line 237, explain what “differntially expressed miRNAs stand for: are them overexpressed or underexpressed in patients respect to healty controls? In lines 260-261, authors again should specify the prognostic value of miR 182 and miR 20 FOR RECURRENCE: is the high expression or the low expression of those miRNAs that is prognostic?